# Researchers' perspective of real-world impact from UK public health research: A qualitative study

**Kay Lakin**⊙*, **Katie Meadmore**⊙, **Alejandra Recio Saucedo, Genevieve Baker**⊙,
**Louise Worswick, Sarah Thomas**⊙

National Institute of Health Research Evaluation, Trials and Studies Coordinating Centre (NETSCC), School of Healthcare Enterprise and Innovation, University of Southampton, Southampton, United Kingdom

* kay.lakin@nihr.ac.uk

## Abstract

Research funded by the National Institute for Health Research Public Health Research Programme is being undertaken in a complex system which brings opportunities and challenges for researchers to maximise the impact of their research. This study seeks to better understand the facilitators, challenges and barriers to research impact and knowledge mobilisation from the perspective of UK public health researchers. A qualitative study using semi-structured interviews, informed by the Payback Framework, with public health researchers who held a research award with the National Institute for Health Research Public Health Research programme up to March 2017 was conducted. Following a thematic analysis, three strongly interlinked themes were extracted from the data and three key factors were highlighted as important for facilitating knowledge mobilisation and impact in UK public health research: (1) Public health researcher's perception of the purpose of the research (2) Approaches to undertaking Knowledge mobilisation activities (3) The complex nature of public health research in the wider research context. These have been reflected onto the Payback framework. Public health researchers can maximise the likelihood for impact by being aware of the context in which they are undertaking research, using different methods, and employing several strategies to take advantage of opportunities. There is a need to support researchers with knowledge mobilisation activities and for funders to identify their expectations of the impact resulting from research. Our findings have relevance to public health researchers and funders interested in increasing the benefit that research brings to society.

## Introduction

The National Institute for Health Research (NIHR) is a major funder of health research in the UK and funds research to improve the health and the wealth of the nation [1]. The NIHR is an important funder of public health research [2], and one route for this is via the Public Health Research (PHR) programme. Established in 2008, the PHR programme funds research to evaluate the effectiveness and cost effectiveness of interventions that take place outside of the

**Data Availability Statement:** As per the ethical approvals in place for this project and to maintain the confidentiality of the interviewees, whole interviews will not be shared. A sample of relevant

quotes are made available in the manuscript under each theme. We are happy to consider requests for additional data, if required. Requests should be made to the Research on Research office based at the University of Southampton via ronr@soton.ac.uk.

**Funding:** This research was funded by the NIHR Evaluation, Trials and Studies Coordinating Centre (NETSCC) https://www.nihr.ac.uk/. The authors work(ed) for NIHR. The views and opinions expressed are those of the author(s), and do not necessarily reflect those of the NIHR, the Department of Health and Social Care, or of NETSCC.

**Competing interests:** All authors worked for the NIHR Evaluation, Trials and Studies Coordinating Centre (NETSCC) at the time of this study. However, no author was involved in any aspect of the research management process or any funding recommendations. This does not alter our adherence to PLOS ONE policies on sharing data and materials.

National Health Service (NHS) to inform delivery and improve the health of the public and reduce health inequalities.

Increased international interest of research funders to understand if and how impact of research takes place is particularly relevant to public health given the multi-sector reach of this research. This interest comes from a shrinkage of the research budget and a need to demonstrate what has happened as a result of public investment on research [3, 4]. For example [5], made recommendations that the NIHR routinely assess the impact of the trials it funds on subsequent systematic reviews and guidelines.

In this paper, research impact means any demonstrable change or effect that has happened in the real world as a result of the research (i.e. outside of the academic environment) [6]. Research impact can be seen as a measure of the effectiveness of knowledge mobilisation (KMb) activities which connect knowledge produced to the 'real world' [7]. Understanding the way in which knowledge is mobilised (i.e., how knowledge is produced, exchanged, disseminated, and translated) to inform policy and practice is important for research teams who are under increasing pressure to undertake activities which seek to maximise the impact of their research. In theoretical evaluations, logic models and frameworks represent, in a simple diagram, the underlying theory of how an intervention(s) leads to outcomes and wider impacts, as well as helping to organise what evidence is needed (indicators) and what data needs to be collected. Logic models have been used by research funders to demonstrate the impact of the research they fund [8] and one of the most widely used models used in evaluating impact from health research is the Payback Framework [8–10].

The interface between researchers and research users plays an important role in evaluating health research impact [10]. Overall, models of how research is adopted into policy show that "adoption of knowledge is interpreted as something that requires partnership working between researchers and policy makers" [11]. However, often it is the researchers alone who shoulder the responsibility for reporting impact, which is not desirable [12]. For instance, it has been proposed that the assessment of research impact (such as the Research Excellence Framework (REF) in the UK [13]) often centralises the role of the academic and overlooks the role of evidence-users [7]. However, the evidence-user perspective is often centralised in studies looking at KMb [14–16] specifically because evidence-users are responsible for implementing research findings into practice. Therefore, there appears to be an unbalanced commitment between those who are responsible for making evidence-informed decisions within the health care system (i.e., policy-makers, practitioners, commissioners) and those (i.e. researchers, evidence producers) who bear the responsibility to demonstrate that knowledge mobilisation of their research informed policy-making decisions and as a result generated impact.

This is especially relevant in Public Health (PH) research. Within the UK, PH research is funded through diverse mechanisms and has many pathways to impact [17]. The effectiveness of these pathways and the impact the research has have been found to depend on a number of factors. For example, contextual factors (e.g. working relationships, values, interpretations of the evidence), researcher roles (e.g. negotiators), collaboration, and KMb outcomes all play a role in the use of evidence within the decision-making process [18]. Similarly, constructs relating to the researcher (expertise, motivation), support (support structure, collaboration, relationship building and understanding needs) and utilising wider dissemination all contribute to generating PH policy impact [19]. Similar approaches from influential Australian public health researchers have been reported [20], who were found to employ a range of tactics to bridge the gap between their research and policy including engaging, targeting, consulting, and forming relationships with key stakeholders throughout the research process. Their findings suggested that the underpinning values of the researcher played a role in what they considered most important for research impact/ KMb. In addition, with regards to KMb,

Australian public health academics considered how they interact with (and value) policy makers and the skills they need to do this, the constraints of their own research environment (context), and the professional identity of the researcher [16]. For curiosity-driven researchers, undertaking quality research and ensuring that the research was published in high impact journals was highly valued. Conversely, 'policy-driven' researchers valued activities which ensured research was influential to policy [20].

Given the importance of KMb and its role in helping to achieve research impact, it is vital that research funders within the UK context, such as NIHR PHR programme, better understand how they can support public health researchers to effectively mobilise research findings to inform public health decisions. This is particularly important because researchers are responsible for reporting their KMb activities and impact(s) achieved to funders and sponsors. Thus, the aim of this paper is to better understand the facilitators, challenges, and barriers to research impact and KMb from the perspective of public health researchers funded by the NIHR PHR programme informed by the Payback Framework.

## Methods

### Design

This study is a qualitative exploration of public health researcher perspectives. It was part of a wider exploratory impact assessment primarily to demonstrate accountability which was undertaken in 2017 on all research awards (n = 113) funded by the NIHR PHR programme between May 2009 and 14<sup>th</sup> March 2017 [Lakin K, Baker G, Thomas S, Worswick L, Dorling H. NIHR Public Health Research Programme: Exploring the influence of research on policy & practice. Unpublished internal report; 2018]. The Payback framework underpinned the wider impact assessment. This framework uses a logic model of the research to create a multi-dimensional characterisation of paybacks resulting from research. This framework was selected as it has been used to evaluate the impact of other NIHR funding research programmes [8], and the PHR Programme had a specific interest in exploring two of the Payback categories, namely Benefits to informing policy, and Benefits to [public] health sector. For this study, public health researchers' experience of the barriers, challenges and facilitators to impact was explored within this context to contribute to a collection of award level case studies building an understanding of the researcher-research user interface within the context of public health. A phenomenological qualitative approach underpinned the study as we were interested in describing and understanding the experiences of our participants. Data were then collected using semi-structured interviews that were informed from the Payback framework (see S1 File).

### Recruitment

A purposive and self-selecting sample was used to identify principal investigators from NIHR PHR programme funded research awards which 1) had demonstrated evidence to suggest that the research findings had been influential to policy and/or the [public] health care sector following a desk-analysis of the PHR portfolio based on the Payback framework [9, 10] and 2) were completed and the findings were published within the NIHR PHR Journal. Research awards which had been influential to policy and/or the [public] health care sector were targeted because this allowed an exploration of proven facilitators and barriers to the potential impacts being realised. Award selection based on completed research was most appropriate given the time-lags that often exist between completing the research and realised influence [21, 22].

From a total of 113 PHR awards available, nine principal investigators from different PHR funded projects were eligible and invited to participate in an interview or to nominate another

member of their research team. Invitations were sent out by email by a member of the project team.

## Ethics approval and consent to participate

Although not standard process within research impact assessments, ethical approval from the Faculty of Medicine University of Southampton Ethics Committee was sought (ID 27460). Study participants were informed about the objectives of the study, any risks and benefits to them, the fact that participation was entirely voluntary and that they could withdraw from the study at any time. Verbal consent was obtained, recorded and stored separately before the start of each interview, as approved in the ethics application. Verbal consent was obtained as this was felt to be the most practical approach given that all interviews were conducted via telephone. Participants were sent a copy of the information sheet and consent form in advance of the arranged interview and were asked to verbally confirm consent. Participant names were anonymised in the interview transcripts. Records linking participant names, consent and transcripts were stored according to the approved protocol.

## Data collection

Semi-structured telephone interviews were undertaken (Aug—Sep 2017) with one interviewer and one note-taker and lasted up to one hour. The interviewer and note-taker undertook the interviews together in a private room at the NIHR Evaluation, Trials and Studies Coordinating Centre in Southampton. The interview framework was adapted from [17], which aligned to the Payback Framework, and sought to understand the nature of the influence or impact the research had realised. However, it also sought to understand how change came about or what stopped it, thereby investigating the way in which knowledge from research was mobilised. The initial interview framework was sent to participants in advance of the interviews, and a question was added to the framework following the first interview to reflect new areas for exploration in further interviews.

## Analysis

Interviews were transcribed by a transcription service and reviewed for accuracy. NVivo 11 qualitative data analysis software was used to support the thematic analysis. Data were analysed according to [23]. Following familiarisation, transcripts were coded. A second coding exercise led to the identification of common themes and sub-themes. Through an iterative process, themes and interpretations were reviewed and discussed by the research team to refine themes through merging, overlapping and separating out key concepts. The coding process was inductive as no prior themes or framework was considered and the themes were data driven. During the analysis, authors were mindful that they work within the NIHR and have a range of expertise in public health, health services research, health evaluation and commissioning, which strengthened the analysis and minimised bias. No one from the project team had a relationship with the participants prior to study commencement. Quotes were reported verbatim. The COREQ guidelines (COnsolidated criteria for REporting Qualitative research) [24] for reporting qualitative data were utilised throughout and used as an additional reporting quality check.

## Consent for publication

All participants consented for their interview data to be used as part of a thematic analysis which would be published. As good practice, a copy of this manuscript was sent to all study participants to inform them of our plan to publish via a journal article and ensure that the interpretation reflected the discussions during the interview.

## Results

Seven researchers representing six research awards were interviewed. This number of interviews was sufficient to provide illustrative examples based on our knowledge of the awards and our experience of qualitative interviews [25]. The research awards covered a range of areas relevant to public health practice (community and education, physical activity/obesity, transport) and research designs (systematic review, randomised controlled trial, natural experiment).

Three over-arching themes, with sub-themes, were extracted from the data using thematic analysis: 1). Public health researcher's perception of the purpose of the research; 2). Approaches to undertaking KMb activities; and 3). The complex nature of public health research in the wider research context.

### 1. Public health researcher's perception of the purpose of the research

This theme suggests that how public health researchers perceive impact and KMb is guided by their views of the purpose of research. There was a variety of views regarding what was considered impact, how to achieve impact and whose responsibility it was to demonstrate impact. Researchers discussed impact and KMb in terms of publications, engagement with the public, policy makers and the media, dissemination, patient and public involvement, changing practice, guidelines or policy, and involvement in the UK REF.

*And we've tried to illustrate, you know, how big an impact we have had, not just in terms of, you know, the number of publications, which has been, yes, much bigger than I think we had anticipated, but also the wider, kind of, policy impact. Participant 4*

These preferences to impact and KMb sat on a continuum that ranged from a fixed pipeline approach to a more integrated, collaborative perspective. Researchers at the pipeline end of the continuum preferred to produce knowledge and evidence via publication that was then available for others to use. This preference tended to give less emphasis to the researcher being involved in how research findings were used in the world after project completion.

*I think my job is to produce some knowledge, and then it's, kind of, not really my skillset to think: well. . . You know, it's also none of my business how that gets used in the world, it's just not what I think I my job is. Participant 1*

By contrast, at the opposite end of the continuum, researchers preferred to undertake their research in a more collaborative and engaging way to bring about change within communities or by influencing changes to policy or practice. These researchers felt that undertaking research was a collective responsibility.

*Participatory research, characterise impact as change that happens during the process as much as what happens beyond it. And if you involve people, [. . .] the impact will go beyond, be more likely to happen because people own the research. So people who have been involved will own it and, therefore, more likely change what they do. Participant 6*

Most researchers seemed aware of their preferences on this continuum. However, they also understood the importance of different activities in facilitating impact and KMb of their research. As such, despite their preferences, researchers still engaged in or employed strategies to ensure different types of activities were undertaken (see theme 2).

Preferences on the continuum were discussed within the context of objectivity and advocacy. It was suggested that the boundary between advocacy and objectivity was sometimes difficult to navigate, with researchers indicating that certain activities aligned them more to one or the other. This could act as a barrier to researchers discussing and planning impact as they sometimes felt uncomfortable attaching themselves to findings or speculating on potential impacts.

*Some researchers, I think, tend to lean more on the advocacy side than the objectivity side, but we've been really careful to try and make sure we can provide evidence when people want it, but. . . we remain objective, not advocates. Participant 4*

## 2. Approaches to undertaking knowledge mobilisation activities

This theme encompassed the strategic approaches that researchers used to facilitate impact and KMb. There were three subthemes (2.1.) strategy and opportunism; (2.2.) enduring networks and partnerships; and (2.3.) role and skills of the team.

**2.1 Strategy and opportunism.** A few participants noted that although they did not have a formal strategy for impact at the start of the research, writing pathways to impact statements as part of funding applications, or the requirements for REF2014, were influential to their thinking about impact and how it can be evidenced. However, challenges were noted around the type of evidence needed.

*One of the major challenges in maintaining and recording, or collating the evidence of impact, is lack of resource. Participant 3*

Researchers reported different impact strategies. One was to give prominence to the research outputs, while another strategy was to plan in advance and to give prominence to having the right people involved.

*I don't believe we can control things or control people, but if you set up practices that encourage people to increase impact. . .then that is where the individuals who are involved [will have an] impact. Participant 6*

However, a few participants noted that the way research impacts the real world can be 'accidental' and therefore a challenge to plan for or predict. Taking advantage of 'windows of opportunity' was viewed positively (see also theme 3). This could involve undertaking planned activities which maximise the chance of encountering opportunities, such as networking, or taking advantage of chance encounters.

*You just happen to meet somebody, a policy maker, at an event. Or, you get invited to a conference from somebody in the audience. And then. . . the actual process is by which it happens are very difficult to plan in advance. Participant 2*

Another strategy was to disseminate and increase the reach of research using media.

*I think, in terms of impact for practice and probably policy, the best way of dissemination was through media: television, radio. . . Yes. I think that's the best. . .. . . it's the high level, perhaps television, sort of, snapshots, that really, really help in terms of getting it out there and getting people to be interested in it. Participant 3*

However, one participant noted that this also came with challenges because the way research findings are presented in the media can mean public reaction is difficult to predict.

*There were some concerns about, kind of. . . ..how the message. . . how our findings might get spun, I guess, in the public domain. There was a kind of concern about what we found, and how that might get utilised. So, I guess it was. . . it's not so much impact, as, kind of, a concern that we'd have to be quite careful with the messaging, in terms of what we found. Participant 1*

All of these challenges indicate that there is a need to approach impact strategies with flexibility.

**2.2 Enduring networks and partnerships.**   All participants agreed that networks and collaborations played a role in achieving impact. Views ranged from *"they were very good"*, to networks and relationships being key to the *"success"* of the research. Most participants noted that enduring relationships are most effective in facilitating impact. As one interviewee stated.

*It's having the time to have those conversations to build those relationships that then means you have a mechanism, going forward, that allows ongoing implementation and ongoing impact to happen. Participant 3*

One participant felt that building relationships was a way to show how they, as a researcher, could be useful to others. Another view was that networks provided a way to raise their profile and research in their field. Some participants felt that engaging with community stakeholders was nice to do but unnecessary. For other participants, local community involvement was a key part of the research process. Almost all participants stated that they engaged with stakeholders, but for some this was is not always easy.

*Policymakers aren't used to really making links with public health researchers. And so they might have different views or opinions about what we should be evaluating. Participant 4*

**2.3 Role and skills of the team.**   Some of the challenges to generate impact within public health were expressed in terms of the skills required to undertake KMb activities, which were felt to be outside of the researcher's area of expertise. In order to overcome this, nearly all participants acknowledged that there is a need to work with people with a range of skills. A number of key skill areas were mentioned: networking, communicating, the right professional background, and the ability to be alert to opportunities.

*What I don't do very well is public engagement or policy impact. And it is worth thinking about hiring somebody who does have those skills, as well. . . rather than just hiring somebody. . . who's really good at academic research. Participant 1*

A couple of participants specifically mentioned the role of knowledge brokers or community members, which were either internal or external to the core research team.

*[They] help us not just with writing press releases but also things like evidence submissions and evidence briefs that we give to policymakers and practitioners Participant 4*

*We wanted the actual end user, if you like, to be part of the process. So for me, impact is. . . it comes from involving people in the process of research. Participant 6*

There was an acknowledgement that having people with a range of skills on a project team is beneficial, particularly when researchers have additional skills which can be useful for the team. However, that wasn't to say that all team members had to have a mixed skill set, and some participants expressed preference to conduct activities close to their research strengths.

*I want the freedom to be able to focus on the academic outputs, and do what I do well.* Participant 1

*There's lots of people out there that are natural brokers, they link people up together* Participant 7

Undertaking research with external stakeholders presented opportunities for skills building. However, a challenge was bringing together team members. Furthermore, some participants indicated that more support and training for researchers is needed to overcome a barrier of effective conversations with non-academic groups or for dissemination activities.

*Anything that could be done to overcome. . . problems that solidify relationships with researchers, academics, policy makers, and practitioners, would be good.* Participant 2

However, irrespective of the team skill-mix, a key barrier to impact was keeping the team together once the funding had finished, or finding capacity to undertake KMb activities during the research.

*The challenge around doing that is that we don't have funding to take the time to write the articles before we finish the work.* Participant 5

### 3. The complex nature of public health research in the wider research context

The nature of public health research is complex. Under this theme there are two subthemes that highlight the facilitators and barriers to KMb and impact delivery within the wider research context: (3.1.) the role of the research funder and (3.2.) the role of wider environmental and societal context. These subthemes interact with the research activities outlined in theme two.

**3.1. The role of the research funder.** Understanding the funder's expectations for impact was discussed by some participants. However, there were concerns around how far researchers should go to plan for impact at application stage.

*I think researchers are a little anxious that if they kind of guess or put some hypothesis for that, they might not be seen very favourably by funders who like to see very prescriptive plans that are all sewn up.* Participant 7

There was a view that funders could do more to support planning for impact, particularly when they commission the research.

*The funders could help by being very clear when they are putting out a request for proposals, that they're anticipating certain types of impact that might come out as a result of the research needs that they see.* Participant 6

Linked to this, there were views on the help and training support that funders could provide to prepare researchers to engage in impact and KMb activities.

*You know, every case is going to be different, and it really is a part of building up your own skills, but at least if you have some sort of training in terms of what you might expect, it would really help going forward. Participant 3.*

Most participants felt that changes need to be made in the funding model to include time for "impact" or dissemination activities. One idea was a pot of funding, a *"sprinkling that continues after the project" (Participant 1)*, that would provide continued capacity once the research has finished, for activities including *"media, television, radio" (Participant 3)*. Some participants reported that they had to secure additional funding or time for these types of activities either as part of their research funding, from their collaborators, or their academic institutions.

There were views about research methodologies that may be more appropriate to public health and the need for funders to take more risks to make sure that the type of research being funded uses the most appropriate methodology. These may be different to the 'gold standard' of randomised control trials, and researchers and funders should be more flexible in their approach to generate evidence in public health spaces. Although it was acknowledged that trials are useful, they are not always feasible and where this is the case, other designs should be used.

*[Funders] should focus on the, sort of, massive things in different centres, which can nudge people in the right direction of making healthier decisions. And often those kinds of interventions won't be trialable because there will be things that we can't control the allocation of them. Participant 2*

Others noted the limitations around systematic reviews in public health, which were described as good but *"messy"* (*Participant 5)*. Some participants also questioned how useful they were to public health practitioners in order to inform practice.

*Health systems are not finding systematic reviews very helpful, to tell them what to do–the practice Participant 6*

In comparison, realist reviews were viewed as a more helpful methodology to address questions within complex systems.

**3.2. The role of wider environmental and societal context.** The wider environmental and societal context in which public health researchers operate was mentioned by some participants. For example, there was an observation around the way in which the health sector uses evidence, in that it is dismissive of or removes evidence which provides context. In this space, the political environment (agenda) was viewed as an uncontrollable influential factor to the impact of public health research.

*By the time it did get funded, it had become quite politically contentious [. . .] there was concern about how findings would be utilised. Participant 1*

However, participants discussed the value of the serendipitous opportunities that influence the impact of public health research. The fast-paced, changing nature of public health arena means that a constant search for windows of opportunities and investment in enduring networks is needed. This is linked to Theme 2.

The mechanism of how research brings about change was one area in which all participants were unified; that the mechanism of change is often through a *"combination of things"*, or as a

result of an increasing body of evidence that *"chips away at the edges" (Participant 1)*. It was acknowledged that unlike other, more clinical research, there is not really the same type of pathway in public health research. Often, research impact in public health can be about changing perceptions, which in turn, adds to a developing *"cultural discourse"* and *"social movement"*.

> *Alongside that evidence, and alongside various other bits of evidence, and other little bits of research I was doing, I fed into, effectively [a] movement amongst academics, and policy makers. Participant 2*

There was an acknowledgement of changes in the way public health services are provided where social enterprises and complex organisations are more involved. The opportunities that this new configuration of public health services present, and the challenges that arise from these were recognised.

> *Commissioners demonstrated a real willingness to have a different sort of relationship with community providers. Participant 6*

Participants identified barriers in the form of differences between practices in academia and the ways in which knowledge spreads in the community, which is on ongoing struggle that some researchers are familiar with. Participants also discussed how the temporary, fixed-term contracts in academia mean that research teams are in a constant state of flux. This runs counterintuitively to the need for long-term commitment required to produce impact in public health.

> *One thing that kind of undermines [. . .] impact is the constant organisational restructuring, and people. Participant 1*

## Linking findings to theory

Our findings highlight areas for consideration for the Payback framework with regards to public health research. Fig 1 highlights that for stages 0–3 of the framework (research topic identification and research activity processes) our findings emphasise the need for and role of the research support community, such as HEIs and researcher funders, to support researchers to plan and undertake KMb activities. Interface B (Dissemination) highlights the need to take advantage of windows of opportunity to inform stage 4 (policy developments), which in turn present opportunities for adoption (stage 5). Networks and partnerships play an important role throughout all stages of the framework and are particularly important for maximising the adoption of research findings within public health contexts. Finally, our findings reflect the researcher perspective in terms of preferences, role and skills and how they can influence if and how they plan for and undertake KMb activities.

## Discussion

The aim of this study was to better understand the facilitators, challenges and barriers to research impact and KMb from public health researchers within the UK context, working outside the NHS setting. Three main themes were extracted from the data. These themes were all interlinked, showing the intricate nature of public health. From these themes, three key factors were highlighted as important for facilitating KMb and impact in public health, which are reflected onto the Payback framework: 1) An awareness of the context is essential in UK public

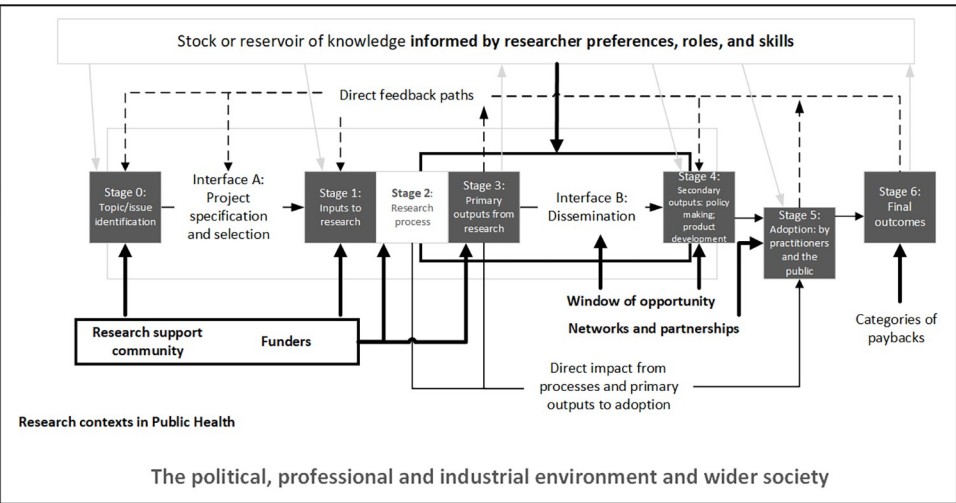

**Fig 1. Study findings mapped onto the Payback Framework Logic Model for assessing research impact.** The Payback Framework Logic Model with areas for consideration resulting from this research highlighted in bold. The framework was adapted from [10].

health research and researchers preferences to KMb activities help navigate this context to achieve impact; 2) seizing on and proactively creating windows of opportunity to engage stakeholders who may be in a position of power in the policy arena or embedded in the community are vital to impact and KMb in public health; 3) funders and research organisations have opportunities to specify their expectations and to take a flexible stance towards impact supporting research teams to navigate the public health context.

## Context is important in public health research

Researchers in public health operate within a complex system where health outcomes are related to a multitude of interdependent environmental, social and physical determinants [26, 27]. Factors such as changes in the political landscape, public opinion and organisational restructures present challenges to the research environment. This means that the context in which a public health researcher works (across third sector, multi-government agencies and private sector) is likely different to other types of health research. Context has been recently highlighted as an area that should receive greater consideration to aid understanding of population health interventions [27, 28]. The importance of context has been reiterated by public health researchers in Australia [16] and UK [18, 27]. Our findings mirror this complexity when researchers discuss the need to navigate context and to ensure their research is relevant. For example, researchers reported that evidence did not always follow a linear evidence-based medicine model and there were concerns raised by participants about how to demonstrate and evidence impact, particularly when it is conceptual (i.e. adds to a body of knowledge) in nature.

Whilst [17] found that researchers questioned the type of evidence needed from public health research to inform policy, our findings suggest that researchers were aware of the type of evidence needed and were using non-standard research designs to ensure that the research generated relevant evidence. Evidence derived from realist reviews, natural experiments, non-randomised trials, novel statistical methods and complex systems approaches provide alternative ways to explore complex interventions [26, 29] and some of these approaches were used by the researchers. This is because in complex social interventions, isolating health outcomes

in order to measure statistically observable effects is difficult and often not the main goal. Moreover, there are practical challenges in trying to randomise service evaluations and population-level interventions, where other forms of evidence maybe better suited [26, 30–32]. Despite this, we found that researchers feel frustrated by the research funding process and the inability to undertake research that is fit for purpose. Indeed, as pointed out in [30], there is a mismatch between the type of evidence required by public health organisations, those who develop evidence-based guidance, and the types of research that funders are willing to support. The findings in [17] and in our paper seem to evidence this mismatch.

Researchers recognised the need to collaborate and/or engage to ensure that the research is positioned within the right context. However, in line with [16], a lack of capacity (skills and time) to do so was a barrier to effective KMb (and therefore impact). Public health researchers need to engage with a broad range of stakeholders who often have different needs, priorities and entrenched practices. Researchers undertook a range of activities to navigate this space, but our findings suggest that more support is needed to help public health researchers with these activities.

## Researcher preferences to knowledge mobilisation activities can facilitate navigation of the public health context

Our findings show that researcher preferences to impact sat on a continuum from a 'traditional academic' to a coproduced approach. This type of continuum has been reported by others [17, 20, 33]. Building on this, we found that the continuum for PH researchers is influenced by views of academic rigour, emphasising objective generation of evidence on one end and co-production and engagement activities on the other. Influential PH researchers in Australia rarely just let research *speak* for itself and instead are aware of the applied nature of PH research [20]. Our findings lend weight to this, and even where a researcher's preference could be viewed as a more traditional academic (i.e. to produce good quality evidence), their research, by its very nature, had policy goals or was policy-driven i.e. *"policy-makers identify and prioritise particular problems that then become the focus of research"*, [28 p. 28]. Similar to [16], public health researchers in this study had a strong awareness of professional identity around researcher role and skill-set, and how they viewed their contribution to society. By contrast, we found no evidence that they viewed undertaking KMb activities as a competitive advantage to their peers. In general, however, our findings suggest that researchers were aware of their preference, which often facilitated how they navigated the complexities of the context in which they work. Such awareness drove different approaches and strategies to impact and KMb. However, as concluded in [16], while researchers in applied research are increasingly expected to engage in KMb activities, this has not been followed by funding or other incentives and rewards which help to overcome barriers. Similar findings have been reported for PH research funded in the UK [17, 27]. One way in which funders could support researchers is to allow for flexible impact plans and strategies in order to respond to the changing complex environment. Moreover, our findings suggest that researchers believe that funders could be clearer on the intended changes or impacts of the research they are funding, and what evidence, data or indicators they need from researchers to demonstrate this change.

## Public health researchers need to seek out windows of opportunity and engage with a wide range of stakeholders

One way in which almost all researchers navigated the public health context was to be aware of windows of opportunity to engage with policy and key stakeholders, and that this was a facilitator to impact. Changes in the policy and health landscape opened up opportunities for

researchers to engage. Such opportunities also presented challenges. Indeed, partnerships can be easier to forge when the policy area is a 'hot topic' and therefore high on the political agenda but this makes sustaining partnerships in long term more challenging [34]. Our findings suggest that researchers were aware of the need to expand the composition of the research team to take this into account including collaborating (and coproducing) with key stakeholders, building on skill sets to engage or building on existing networks and partnerships. As found in [18], researchers evaluating public health interventions often had to change/shift their role depending on the context. A systematic review of KMb in the third sector found that strong relationships with academia were important to the mobilisation of research evidence and that having the ability to coproduce research could help to overcome some of the barriers by ensuring that the research was applicable to a local context [35]. Other strategies included using local or national research brokers. Knowledge brokers are being increasingly used to facilitate KMb between the producers and users of evidence to overcome the struggles of mobilising research knowledge into policy and practice. A number of different aims for knowledge brokers have been identified to facilitate KMb [36]. Mobilising knowledge to inform decision-making takes time but an important aspect for a broker is the ability to respond to windows of opportunity (e.g. hot topics, capitalising on chance encounters and creating networking opportunities) while at the same time hold evidence to inform future policies [34]. Despite such strategies, our findings are in line with others [17, 27] and suggest that there is a need to provide more support for researchers such as the provision of training and building of skills to help with the range of stakeholders that relationships need to be forged with and the different ways in which the research is communicated to different audiences.

## What this study adds

The recent yet increasing interest around impact to measure research performance is presenting challenges to stakeholders involved in research. This study contributes to the field by shedding light on challenges PH researchers in the UK are struggling with in defining, generating and demonstrating impact, and more importantly, what actions could be taken to develop the skills needed to track and capture impact from research.

This work is presented from the point of view of the PH researchers and provides a snapshot of the landscape of how impact operates in this context. Recommendations arising from the researcher's perspective here suggest that funders and research organisations should consider how best to support research teams. Such support could include specifying expectations towards impact, being open to the different methodologies that researchers in this field can adopt, provision of funding to support and enable KMb activities, committing to develop the skills that researchers need to engage with a wide range of stakeholders, and to evaluate track and capture impact from their research. This study has been laid out as a platform to encourage further work in the field. To develop a full picture of the views and interactions of all stakeholders, next steps would be to compare the views here with those from evidence-users as well as those who receive the benefits of research, to investigate whether impact is perceived in the same way by all stakeholders. Future research could consider the role of non-academic public health professionals working as knowledge brokers or as part of research team in increasing the impact of research.

Our findings highlight areas for consideration on the Payback framework specifically in the context of researchers undertaking public health research. Our findings have been mapped onto the framework emphasising the additional elements researchers should consider when planning for impact. As reflected in the model, our findings suggest that while the interface between researchers and research users is important, this is more complex than a two-way

interface. The role of the research funder and the research institution, is important in this space to ensure that the right risks are being taken to fund the right research, with a team with the right skills in KMb (and provide training and support to achieve this). This will be of interest to anyone who is undertaking a theory driven impact evaluation or assessment of public health research to support what factors could influence the resulting outcomes and impacts of the research.

Since the introduction of the PHR programme, responsibilities for commissioning and providing public health services in England were transferred from the NHS to local authorities [37]. Although such transfers of responsibility may not have resulted in changes to the research environment for the researchers interviewed in this study, it is possible that it may open up new opportunities to engage in evidence-based policy-making, which is applicable to public health [38]. The number of interviews was small, and it is possible that a larger sample size might yield additional findings. We only looked at research which had completed but we acknowledge that impact may have resulted from active research. The research was undertaken from the perspective of the NIHR; participants were chosen from studies funded by the NIHR only and other funders might have different support provision for researchers in public health programmes. In addition, the authors work within the NIHR and although our selection of quotes gave a broad range of opinions, we do not know whether this factor might have influenced the openness of the participant responses. Our findings relate to public health research, specifically research which is undertaken outside of the NHS, and may not be generalisable to public health research conducted in NHS settings.

## Conclusion

The findings in this study are largely consistent with the existing research examining the views of public health researchers on impact and KMb. However, its contribution provides better clarity on public health researchers within the UK context, and in particular, on research being undertaken outside the NHS setting and this has been reflected in the Payback framework. Our findings support evidence in the literature which increasingly shows that context is important for the intervention/policy being evaluated. We would add that context is also highly relevant for the research environment and that the skills and approaches of researchers to navigate this context are important. Researchers need to be aware of their preferences for engaging with stakeholders and KMb in order to devise impact strategies that are more likely to succeed. Our findings show that by being aware of the context in which they are undertaking the research, using different research methods, and employing the right strategies to take advantage of opportunities, the researchers were able to maximise the likelihood for change to occur, and any change could be reported back to the research team through the enduring relationship between the parties.

## Supporting information

**S1 File. Interview framework.** Semi-structures questions that were asked of participants during telephone interviews.
(DOCX)

## Acknowledgments

We would like to thank DP, HW and CK for reading a draft of this manuscript and providing constructive feedback. We would also like to say thank you to all participants for consenting and giving up their time to be interviewed in this study, and for providing feedback on the

draft manuscript. Thank you to all our peer reviewers for providing helpful feedback during the ethics approval process and on this paper, your contributions made this stronger piece of research.

## Author Contributions

**Conceptualization:** Kay Lakin, Genevieve Baker, Sarah Thomas.

**Data curation:** Kay Lakin, Genevieve Baker, Louise Worswick, Sarah Thomas.

**Formal analysis:** Kay Lakin, Katie Meadmore, Alejandra Recio Saucedo, Genevieve Baker, Louise Worswick.

**Investigation:** Kay Lakin, Genevieve Baker, Louise Worswick, Sarah Thomas.

**Methodology:** Kay Lakin, Louise Worswick.

**Project administration:** Kay Lakin, Genevieve Baker.

**Supervision:** Sarah Thomas.

**Validation:** Katie Meadmore, Alejandra Recio Saucedo.

**Writing – original draft:** Kay Lakin, Katie Meadmore, Alejandra Recio Saucedo, Louise Worswick, Sarah Thomas.

**Writing – review & editing:** Kay Lakin, Katie Meadmore, Alejandra Recio Saucedo, Genevieve Baker, Sarah Thomas.

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
