## [Decision Letter · Decision Letter 0]

23 Sep 2021

PONE-D-21-03495Researchers’ perspective of real-world impact from UK public health research: a qualitative studyPLOS ONE

Dear Dr. Lakin,

Thank you for submitting your manuscript to PLOS ONE. After careful consideration, we feel that it has merit but does not fully meet PLOS ONE’s publication criteria as it currently stands. Therefore, we invite you to submit a revised version of the manuscript that addresses the points raised during the review process.

We look forward to receiving your revised manuscript.

Kind regards,

Chaisiri Angkurawaranon

Academic Editor

PLOS ONE

Journal Requirements:

2. We note that you obtained audio consent. Please state in the Methods:

- Why written consent could not be obtained

- Whether the Institutional Review Board (IRB) approved use of oral consent

For more information, please see our guidelines for human subjects research: https://journals.plos.org/plosone/s/submission-guidelines#loc-human-subjects-research

3. In your Methods section, please provide additional information about the participant recruitment method and the demographic details of your participants. Please ensure you have provided sufficient details to replicate the analyses such as descriptions of where participants were recruited and where the research took place.

4. Please include a copy of the interview guide used in the study, in both the original language and English, as Supporting Information, or include a citation if it has been published previously.

5. Thank you for stating the following in the Competing Interests section: "All authors worked for the NIHR Evaluation, Trials and Studies Coordinating Centre (NETSCC) at the time of this study. However, no author was involved in any aspect of the research management process or any funding recommendations."

6. We note that you have indicated that data from this study are available upon request. PLOS only allows data to be available upon request if there are legal or ethical restrictions on sharing data publicly. For more information on unacceptable data access restrictions, please see http://journals.plos.org/plosone/s/data-availability#loc-unacceptable-data-access-restrictions. 

7. Your abstract cannot contain citations. Please only include citations in the body text of the manuscript, and ensure that they remain in ascending numerical order on first mention.

8. We note that you have referenced "Lakin K, Baker G, Thomas S, Worswick L, Dorling H. NIHR Public Health Research Programme: Exploring the influence of research on policy & practice" which has currently not yet been accepted for publication. Please remove this from your References and amend this to state in the body of your manuscript: "Lakin K, Baker G, Thomas S, Worswick L, Dorling H. NIHR Public Health Research Programme: Exploring the influence of research on policy & practice. Unpublished internal report; 2018." as detailed online in our guide for authors http://journals.plos.org/plosone/s/submission-guidelines#loc-reference-style.

Reviewers' comments:

Reviewer's Responses to Questions

**Comments to the Author**

1. Is the manuscript technically sound, and do the data support the conclusions?

Reviewer #1: Yes

Reviewer #2: Partly

2. Has the statistical analysis been performed appropriately and rigorously? 

Reviewer #1: Yes

Reviewer #2: N/A

3. Have the authors made all data underlying the findings in their manuscript fully available?

Reviewer #1: Yes

Reviewer #2: Yes

4. Is the manuscript presented in an intelligible fashion and written in standard English?

Reviewer #1: Yes

Reviewer #2: Yes

5. Review Comments to the Author

Reviewer #1: There are no major suggestions as such. The manuscript is well written and meets the standard criteria and has passed the COREQ checklist successfully meeting the standards and encompassing all the necessary aspects.

However, it is advised that the manuscript is structured accordingly to the criteria highlighted by the journal, although very few minor adjustments are required and correction of few grammatical errors. A professional editing is also advised for typos etc.

Reviewer #2: Nicely done empirical study - I have no major criticisms of that aspect. but it's under-theorised. I think given the HUGE knowledge base about knowledge translation and research impact that's accumulated in the past few years, we can't have more papers appearing which just present "three linked themes". You need to theorise this properly, using one or other of the many theoretical frameworks linking research to impact! Personally I'd use either the CIHR framework or Jo Rycroft-Malone's realist evaluation framework (might be diplomatic - she's now head of HS&DR!). This will mean a major re-write but I think worth it. My team reviewed the different impact frameworks a few years ago - there may be better ones out there now.

Trish Greenhalgh

Canadian Academy of Health Sciences: Making an Impact, A Preferred Framework and Indicators to Measure Returns on Investment in Health Research. Downloadable from http://www.cahs-acss.ca/wp-content/ uploads/2011/09/ROI_FullReport.pdf. Ottawa: CAHS; 2009.

Rycroft-Malone J, Burton C, Wilkinson J, Harvey G, McCormack B, Baker R,

Dopson S, Graham I, Staniszewska S, Thompson C et al: Health Services and Delivery Research. In: Collective action for knowledge mobilisation: a realist evaluation of the Collaborations for Leadership in Applied Health Research and Care. Volume 3, edn. Southampton (UK): NIHR Journals Library.; 2015: 44.

Greenhalgh et al. Research impact - a narrative review. BMC Medicine (2016) 14:78

6. PLOS authors have the option to publish the peer review history of their article (what does this mean?). If published, this will include your full peer review and any attached files.

Reviewer #1: No

Reviewer #2: **Yes: **Trisha Greenhalgh

---

## [Author Response · Author response to Decision Letter 0]

22 Mar 2022

Reviewer #1: There are no major suggestions as such. The manuscript is well written and meets the standard criteria and has passed the COREQ checklist successfully meeting the standards and encompassing all the necessary aspects. However, it is advised that the manuscript is structured accordingly to the criteria highlighted by the journal, although very few minor adjustments are required and correction of few grammatical errors. A professional editing is also advised for typos etc.

We thank the reviewer for these positive comments. We have revisited the manuscript guidelines and revised the manuscript accordingly. We have also had the manuscript proof-read before resubmission. 

Reviewer #2: Nicely done empirical study - I have no major criticisms of that aspect. but it's under-theorised. I think given the HUGE knowledge base about knowledge translation and research impact that's accumulated in the past few years, we can't have more papers appearing which just present "three linked themes". You need to theorise this properly, using one or other of the many theoretical frameworks linking research to impact! Personally I'd use either the CIHR framework or Jo Rycroft-Malone's realist evaluation framework (might be diplomatic - she's now head of HS&DR!). This will mean a major re-write but I think worth it. My team reviewed the different impact frameworks a few years ago - there may be better ones out there now.

Trish Greenhalgh

Canadian Academy of Health Sciences: Making an Impact, A Preferred Framework and Indicators to Measure Returns on Investment in Health Research. Downloadable from http://www.cahs-acss.ca/wp-content/ uploads/2011/09/ROI_FullReport.pdf. Ottawa: CAHS; 2009.

Rycroft-Malone J, Burton C, Wilkinson J, Harvey G, McCormack B, Baker R,

Dopson S, Graham I, Staniszewska S, Thompson C et al: Health Services and Delivery Research. In: Collective action for knowledge mobilisation: a realist evaluation of the Collaborations for Leadership in Applied Health Research and Care. Volume 3, edn. Southampton (UK): NIHR Journals Library.; 2015: 44.

Greenhalgh et al. Research impact - a narrative review. BMC Medicine (2016) 14:78

We thank Professor Greenhalgh for her comments. We accept that the paper did not explicitly reference the theory behind the thematic framework. The interview questions themselves were informed by the Payback Framework. As such, we have continued to use this theory to link to our findings. We have added a diagram (Fig 1) to show how the findings from this study support and map onto the Payback Framework for PH research and have revised the manuscript throughout to reflect this. We agree that this now significantly strengthens the findings of the paper.

---

## [Decision Letter · Decision Letter 1]

6 May 2022

Researchers’ perspective of real-world impact from UK public health research: a qualitative study

PONE-D-21-03495R1

Dear Dr. Lakin,

We’re pleased to inform you that your manuscript has been judged scientifically suitable for publication and will be formally accepted for publication once it meets all outstanding technical requirements.

Kind regards,

Chaisiri Angkurawaranon

Academic Editor

PLOS ONE

Additional Editor Comments (optional):

Reviewers' comments:

Reviewer's Responses to Questions

**Comments to the Author**

1. If the authors have adequately addressed your comments raised in a previous round of review and you feel that this manuscript is now acceptable for publication, you may indicate that here to bypass the “Comments to the Author” section, enter your conflict of interest statement in the “Confidential to Editor” section, and submit your "Accept" recommendation.

Reviewer #2: All comments have been addressed

2. Is the manuscript technically sound, and do the data support the conclusions?

Reviewer #2: Yes

3. Has the statistical analysis been performed appropriately and rigorously? 

Reviewer #2: N/A

4. Have the authors made all data underlying the findings in their manuscript fully available?

Reviewer #2: Yes

5. Is the manuscript presented in an intelligible fashion and written in standard English?

Reviewer #2: Yes

6. Review Comments to the Author

Reviewer #2: Good response to my comments. They have now highlighted the theoretical framework they used. Happy for publication.

7. PLOS authors have the option to publish the peer review history of their article (what does this mean?). If published, this will include your full peer review and any attached files.

Reviewer #2: **Yes: **Trisha Greenhalgh

---

## [Editor Report · Acceptance letter]

17 Jun 2022

PONE-D-21-03495R1 

Researchers’ perspective of real-world impact from UK public health research: a qualitative study 

Dear Dr. Lakin:

I'm pleased to inform you that your manuscript has been deemed suitable for publication in PLOS ONE. Congratulations! Your manuscript is now with our production department. 

Kind regards, 

on behalf of

Dr. Chaisiri Angkurawaranon 

Academic Editor

PLOS ONE